# Human Hemoglobin and Antipsychotics Clozapine, Ziprasidone and Sertindole: Friends or Foes?

**DOI:** 10.3390/ijms24108921

**Published:** 2023-05-17

**Authors:** Lena Platanić Arizanović, Nikola Gligorijević, Ilija Cvijetić, Aleksandar Mijatović, Maja Krstić Ristivojević, Simeon Minić, Aleksandra Nikolić Kokić, Čedo Miljević, Milan Nikolić

**Affiliations:** 1University of Belgrade-Faculty of Chemistry, Studentski trg 12-16, 11000 Belgrade, Serbiailija@chem.bg.ac.rs (I.C.); krstic_maja@chem.bg.ac.rs (M.K.R.); sminic@chem.bg.ac.rs (S.M.); 2Institute of Chemistry, Technology, and Metallurgy, National Institute of the Republic of Serbia, University of Belgrade, Njegoševa 12, 11000 Belgrade, Serbia; nikola.gligorijevic@ihtm.bg.ac.rs; 3Faculty of Mining and Geology, University of Belgrade, Đušina 7, 11000 Belgrade, Serbia; aleksandar.mijatovic@rgf.bg.ac.rs; 4Institute for Biological Research “Siniša Stanković”, National Institute of the Republic of Serbia, University of Belgrade, Bulevar despota Stefana 142, 11000 Belgrade, Serbia; san@ibiss.bg.ac.rs; 5Institute of Mental Health, Palmotićeva 37, 11000 Belgrade, Serbia; cedo.miljevic35@gmail.com

**Keywords:** antipsychotics, human hemoglobin, clozapine, ziprasidone, sertindole, binding, interactions

## Abstract

Packed with hemoglobin, an essential protein for oxygen transport, human erythrocytes are a suitable model system for testing the pleiotropic effects of lipophilic drugs. Our study investigated the interaction between antipsychotic drugs clozapine, ziprasidone, sertindole, and human hemoglobin under simulated physiological conditions. Analysis of protein fluorescence quenching at different temperatures and data obtained from the van’t Hoff diagram and molecular docking indicate that the interactions are static and that the tetrameric human hemoglobin has one binding site for all drugs in the central cavity near αβ interfaces and is dominantly mediated through hydrophobic forces. The association constants were lower-moderate strength (~10^4^ M^−1^), the highest observed for clozapine (2.2 × 10^4^ M^−1^ at 25 °C). The clozapine binding showed “friendly” effects: increased α-helical content, a higher melting point, and protein protection from free radical-mediated oxidation. On the other hand, bound ziprasidone and sertindole had a slightly pro-oxidative effect, increasing ferrihemoglobin content, a possible “foe”. Since the interaction of proteins with drugs plays a vital role in their pharmacokinetic and pharmacodynamic properties, the physiological significance of the obtained findings is briefly discussed.

## 1. Introduction

Human hemoglobin (HHb), the most abundant blood protein, is an essential respiratory protein in vertebrates’ erythrocytes (red blood cells). Structurally, it’s a tetramer of two α- and two β-polypeptide globin chains, each bound to a redox iron heme in the crevices at the subunit exterior. The α-subunit contains 141 amino acid residues whereas the β-subunit comprises 146 amino acid residues, with seven and eight helical regions, respectively. Each heme group has a porphyrin ring and a ferrous atom capable of reversibly binding one oxygen molecule [1].

Antipsychotic medications are the mainstay for treating the symptoms of schizophrenia and other psychotic disorders, altering brain chemistry to help reduce psychotic symptoms like hallucinations, delusions, and disordered thinking [2]. Three second-generation (atypical) antipsychotics, acting as serotonin-dopamine antagonists, are particularly interesting due to their unique pharmacological properties or controversies associated with clinical use. Clozapine (Figure 1A) is the “gold-standard” antipsychotic for treatment-resistant schizophrenia. Despite its robust efficacy, there is evidence that clozapine is underutilized, partly due to concerns about its potential adverse effects [3]. Although it may be a slightly less efficacious antipsychotic drug, the main advantage of ziprasidone (Figure 1B) is its low propensity to induce weight gain and associated adverse effects [4,5]. Sertindole (Figure 1C) was withdrawn from the market due to concerns about its association with the prolongation of QT intervals, severe cardiac arrhythmias, and sudden cardiac death associated with its use. After re-evaluating its risks and benefits, it is reintroduced into the European market [6]. Still, the US Food and Drug Administration has not re-approved sertindole.

Various factors limit the effectiveness of antipsychotic drugs. Except for the most obvious accompanying adverse effects, the pleiotropic effects of drugs include interactions with biomolecules that are not the intended target of their action. Thus, knowledge of the binding characteristics of therapeutically important molecules is critical for understanding their availability in the requested tissues and their impact on the structure, activity, and function of the macromolecule partner(s). The erythrocyte compartment of the blood is often dismissed as an insignificant consideration in pharmacokinetics. However, many drugs accumulate significantly in human erythrocytes (e.g., [7]), cells that constitute a unique drug delivery system [8]. Mature erythrocytes act as “hemoglobin sacs”, making this transport protein the first target for binding xenobiotics. Indeed, since the pioneering works of Nobel laureate Max Perutz, numerous studies have examined the binding of different exogenous substances to the HHb, including some drugs (e.g., [9]) and food antioxidants [10]. Therefore, this study aims to investigate for the first time the in vitro binding of antipsychotics clozapine, ziprasidone, and sertindole to (commercial and isolated) human hemoglobin (HHb) by various spectroscopic methods backed by molecular docking analysis. The influence of drug binding on hemoglobin’s structure and thermal and oxidative stability was also determined. Findings are significant in assessing the pleiotropic effects of the tested antipsychotics (“friends or foes”) on this miraculously vivid red human protein.

## 2. Results and Discussion

In the present study, we first explored the binding of the atypical antipsychotics clozapine, ziprasidone, and sertindole to HHb under simulated physiological conditions via steady-state fluorescence measurements combined with in-silico docking studies. Poor solubility of drugs in aqueous buffer solutions disables the use of isothermal titration calorimetry for biomolecular interaction analysis. Then, we studied the consequences of ligand binding, evaluating its influence on the α-helical structure and thermal stability of the protein by standard spectroscopic methods. Interestingly, the authors used commercial HHb preparations in most published papers to test the protein binding potential of various xenobiotics [10]. Therefore, the purchased HHb preparation (Section 3.1) was utilized to better compare our findings with the literature, i.e., the interaction of the HHb–various drugs. The exception was examining the oxidative stability of this erythrocyte hemoprotein in the presence of antipsychotics, for which freshly isolated, therefore completely native, oxy-HHb molecules were the starting material.

### 2.1. Fluorescence Quenching of Human Hemoglobin by Atypical Antipsychotics

Quenching studies of protein fluorescence by ligands is a convenient means for exploring ligand-protein interactions. The hemoglobin tetramer is assembled from two αβ dimers. Each dimer contains three tryptophan (Trp) residues: α-Trp14 and β-Trp15 are outside the subunit interface, while the β-Trp37 residue, located at the α_1_β_1_ interface, has been regarded as the primary source of HHb fluorescence emission [11]. Hence, a study of the change in intrinsic fluorescence of HHb in the presence of the antipsychotics may give information about the local environment of the Trp moiety during interaction. Moreover, there are also five tyrosine (Tyr) residues in each αβ dimer: α-Tyr24, α-Tyr42, α-Tyr140, β-Tyr34, and β-Tyr144 [12].

Figure 2A, Figure 3A and Figure 4A present spectra showing the regular quenching of HHb fluorescence in the presence of increasing concentrations of the examined drugs. The working concentration of protein was 4 µM to prevent the quenching of Trp fluorescence by the neighboring heme group [13]. Human hemoglobin exhibited a strong fluorescence emission with a peak at 344 nm on excitation at 280 nm, and all ligands had comparatively low fluorescence compared with the protein. On the other hand, ziprasidone (Figure 3A) and sertindole (Figure 4A) showed strong fluorescence with a peak at ~400 nm, resulting in negative peaks in these areas of the collective emission spectra when sample mixtures were excited at 280 nm. In the case of clozapine and ziprasidone mixtures with HHb, no shift of the peak maximum was observed, while a blue shift of the peak maximum by 8 nm was observed with sertindole. The results suggest that the antipsychotics interact with HHb and the fluorescence quenching was due to specific complex formation, and that, since the β-37Trp is primarily responsible for the intrinsic fluorescence of HHb, changes in protein fluorescence intensity could indicate the presence of a binding site for antipsychotics near β-37 Trp residue [14].

The Stern-Volmer plots of the HHb–antipsychotic systems at different temperatures were essentially linear for drug concentrations up to 28 μM (Figure 2B, Figure 3B and Figure 4B), indicating that only one quenching type (static or dynamic) occurs. In static quenching, the complex formation occurs in the ground state. In contrast, dynamic quenching results from complex formation between the ligand and the protein in the excited state of the fluorophore. It was observed that *K*_SV_ values were on the order of 10^3^ M^−1^ (Table 1), meaning there was a decrease in HHb fluorescence in the presence of the antipsychotic ligands. Based on Table 1 data and the variations of the *K*_SV_ values as a function of temperature, it was possible to characterize the preferential quenching mechanism for all three drugs as static since the values of the Stern-Volmer constants decreased with increasing temperature, thus forming a nonfluorescent supramolecular complex in the ground state [15].

The calculated value of *K*_q_ for the three antipsychotics binding is of the order of 10^11^ L mol^−1^ s^−1^ (Table 1), exceeding the value of 2 × 10^10^ L mol^−1^ s^−1^ acknowledged as the maximum *K*_q_ in the dynamic quenching of proteins [16], a further unequivocal proof that the described fluorescence quenching of HHb is not controlled by diffusion and does not originate from dynamic quenching. In all recent studies [14,17,18,19,20], the binding of the tested drugs to the commercial human hemoglobin molecules was characterized as static.

### 2.2. Binding Constants Determination in Human Hemoglobin and Antipsychotic Systems

Parameters that characterize the interaction between a protein and its ligands are its binding strength and the number of ligand binding sites on the protein. Treating Equation (6) as the equation for a straight line, the value for the binding constant “*K*_a_” and the number of binding sites “n” for the binding of clozapine, ziprasidone, and sertindole with HHb within the studied concentration range of the antipsychotics were obtained from the intercept and slope of the double log plot, respectively (Figure 2C, Figure 3C and Figure 4C).

The decreasing trend of *K*_a_ with increasing temperature for all ligands, although the differences are relatively minor (Table 2) and follow the dependence of *K*_SV_ on temperature, is consistent with the static quenching mechanism mentioned above. Most ligands are bound reversibly and display moderate affinities for the protein partner, e.g., binding constants are 1–15 × 10^4^ M^−1^ [10,21]. Therefore, the *K*_a_ values reported in this paper indicate that the binding between clozapine (1.3 × 10^4^ M^−1^ at 37 °C) or sertindole (1.6 × 10^4^ M^−1^ at 37 °C) and HHb was lower-moderate and that these medications can be stored and carried by the hemoglobin in the human body, given the high concentration of this transport protein in the red blood cells of approximately 330 mg/mL [22]. In comparison with other drugs, these association constants are comparable to those of the anti-mitotic chemotherapy agent docetaxel (3.2 × 10^4^ M^−1^ at 37 °C) [17], the synthetic analgesic tramadol (1.2 × 10^4^ M^−1^ at 37 °C) [18], and the antimalarial chloroquine (1.6 × 10^4^ M^−1^ at 37 °C) [19], but lower than those of commercial hemoglobin and the nonsteroidal anti-inflammatory medication diclofenac (5.0 × 10^5^ M^−1^ at 37 °C) [14]. The estimated association constant for ziprasidone binding to HHb was found to be consistently lower, of the order of magnitude only 10^3^ M^−1^ (Table 2), similar to the value of the analgesic and antipyretic drug acetaminophen (6.4 × 10^3^ M^−1^ at 37 °C), obtained by calorimetry with isolated protein [23], but still higher than that recorded for promethazine (2.1 × 10^2^ M^−1^ at 37 °C) and adiphenine (3.0 × 10^2^ M^−1^ at 37 °C) amphiphilic drugs [20].

The value of n is nearly equal to 1 in all three systems and temperatures (Table 2), suggesting one binding site for these antipsychotics around the Trp residue (s) of the protein, identical to all the studies of interactions of human hemoglobin with drugs mentioned in the previous paragraph.

### 2.3. Synchronous Fluorescence Measurements in Human Hemoglobin and Antipsychotic Systems

Synchronous fluorescence spectroscopy gives information about the molecular environment near a protein fluorophore (Trp and Tyr). It involves simultaneous scanning of the excitation and emission monochromators while maintaining a constant wavelength interval (Δλ) between them. The emission maximum shift reflects the polarity changes around the fluorophore molecule [24]. The synchronous HHb fluorescence spectra with various amounts of antipsychotics are shown in Figure 2D, Figure 3D and Figure 4D (Δλ stabilized at 60 nm, giving characteristic information of Trp residues) and Figure 2E, Figure 3E and Figure 4E (Δλ of 15 nm for Tyr residues). The fluorescence intensity of HHb regularly decreased with the addition of all antipsychotics, further demonstrating the occurrence of fluorescence quenching in the binding process.

Expectedly, the contribution of Trp fluorophores (the signal intensity) to the internal fluorescence of HHb was greater. However, the contribution of fluorophores to protein signal quenching (the relative decrease of fluorescence intensity) in the presence of antipsychotics was different. The clozapine molecules affect the fluorescence intensity of Tyr more than that of Trp (Figure 2E vs. Figure 2D). In contrast, there is a greater decrease in the signal intensity of Trp residues as compared to Tyr with sertindole (Figure 4D vs. Figure 4E). Finally, it seems the fluorescence quenching of HHb comparably originated from both Trp and Tyr residues in the case of ziprasidone binding (Figure 3D vs. Figure 3E). The described results indicate that the binding site of these drugs on the hemoglobin molecule could be different.

It can also be seen that the maximum emission wavelength of Tyr residues in the HHb complex with clozapine (Figure 2E), ziprasidone (Figure 3E), and sertindole (Figure 4E) does not have a significant shift over the investigated concentration range, which indicates that these drugs binding does not affect the microenvironment of Tyr residues in HHb. Likewise, the environment of the Trp fluorophores appears to remain the same after clozapine (Figure 2D) and ziprasidone (Figure 3D) bind to HHb. In contrast, a blue shift (from 347 to 340 nm) of Trp residues was observed in the HHb–sertindole complex (Figure 4D), indicating the hydrophobicity of (at least one) Trp residue increased and this residue was moved to a more hydrophobic environment. Therefore, it seems the conformation of the HHb molecule was changed to some extent upon sertindole binding.

### 2.4. Thermodynamic Study of Antipsychotics Binding to Human Hemoglobin

The sign of thermodynamic parameters such as Gibbs free energy (ΔG), enthalpy change (ΔH), and entropy change (ΔS) involved in protein-ligand binding is helpful for the determination of the type of interaction involved between HHb and the antipsychotics analyzed. Therefore, the quenching experiments have been done at three different temperatures to determine the binding thermodynamic parameters. From the slope (the enthalpy) and intercept (the entropy) of the linear van’t Hoff plots (ln *K*_a_ vs. 1/T; Figure 2F, Figure 3F and Figure 4F) and Equations (7) and (8), these parameters have been calculated and listed in Table 2.

In all cases, the negative sign of the Δ*G* value indicates the spontaneity of the molecular recognition process at all temperatures [25]. The values of Δ*H* were all negative (−7.4 kJ mol^−1^ for sertindole, −9.8 kJ mol^−1^ for ziprasidone, and −12.5 kJ mol^−1^ for clozapine), and Δ*S* were all positive (+41.0 J mol^−1^ K^−1^ for clozapine, +41.2 J mol^−1^ K^−1^ for ziprasidone, and +56.8 J mol^−1^ K^−1^ for sertindole) for all three HHb–ligand systems. The negative values of enthalpy changes (Δ*H*) and positive values of entropy contribution (TΔ*S*) strongly suggest that it is an entropy-driven spontaneous and exothermic binding process. Moreover, the positive entropy changes also indicate that the randomness around the HHb-antipsychotic complexes increases, again evidence of hydrophobic interaction [23]. The negative enthalpy changes may also support the existence of hydrogen bonds in the complexation [25]. Indeed, these assumptions were supported by the docking study results (next section) and the high lipophilicity of these drugs (log P > 3; https://go.drugbank.com/ (accessed on 11 April 2023).

### 2.5. Docking Results

Unlike human serum albumin, the abundant transport protein for non-polar drugs through the bloodstream, which has at least two structurally well-defined major ligand binding sites [26], only the solvent-accessible central cavity of HHb contains functionally important binding sites for several classes of allosteric effectors that facilitate the lowering of oxygen affinity [9], within which various xenobiotics are also bound. According to HHb binding sites, the central cavity can be classified into two subsites: I, close to the corner of α_2_ and β_1_ subunits, may be suitable to bind electro-negative drugs, and II, close to the α_1_ subunit, tends to bind electropositive drugs [27].

The molecular docking technique is frequently used to understand the interactions between proteins and ligands. The results presented in Figure 5 provide insights into the binding interactions of three antipsychotic drugs with the central binding site of HHb. The docking studies revealed that all three drugs bind within the HHb cavity, with clozapine and sertindole occupying similar binding pockets on the α_1_β_1_ interface, while ziprasidone was preferentially placed near the α_2_ and β_2_ subunits (Figure 5A). The mean binding affinities of ziprasidone and sertindole were similar (−8.61 vs. −8.57 kcal mol^−1^), while clozapine exhibited a somewhat lower binding affinity (−7.85 kcal mol^−1^). Unsurprisingly, molecular docking has given the opposite result compared to fluorimetry since docking affinities do not include the entropic contributions; hence, they align with the binding enthalpies obtained from the fluorescence experiments.

Further analysis of the binding interactions revealed that sertindole forms several favorable hydrophobic interactions with specific amino acid residues such as α_1_-Ala130, α_2_-Tyr140, α_1_-Val1, and β_1_-Trp37 (Figure 5B). Additionally, the fluorobenzene ring of sertindole interacts with the positively charged α_2_-Arg141 side chain. Also, the nitrogen atom from the piperidine ring forms electrostatic interactions with α_1_-Asp126, while the carbonyl O atom forms hydrogen bond interactions with α_1_-Lys99. The coplanarity between the fluorobenzene ring of the ligand and β_1_-Trp37 indole ring, along with the relatively short distance between their centroids (5.7 Å), suggests the probability for π-π interactions that would complement hydrophobic interactions between these two moieties. These findings agree with the observed blue shift in the Trp synchronous HHb fluorescence spectrum upon sertindole binding (Figure 4D). On the other hand, ziprasidone preferably interacts with β_2_-Trp37, α_1_-Thr137, and α_1_-Tyr140 through the chlorine substituent on the dihydrooxoindol ring (Figure 5C). It also forms favorable electrostatic interactions between the piperazine nitrogen atom and the carboxylic group of α_2_-Asp126. In contrast, clozapine interacts with α_1_-Ala130, β_1_-Asn108, and α_1_-Asp126 via hydrogen bonding, hydrophobic, and electrostatic interactions (Figure 5D).

Our findings, in agreement with thermodynamic and synchronous results, provide valuable insights into the binding mechanisms of sertindole, ziprasidone, and clozapine drugs within the central cavity of the HHb tetramer. Namely, the docking study confirmed that the hydrophobic effect followed by hydrogen bonds or electrostatic interactions is essential in the binding between HHb and the antipsychotics examined. The drugs docetaxel [17] and diclofenac [14] have also been suggested to bind to the central cavity of HHb, close to subunit α_1_, and the possible binding site of acetaminophen is near β-Trp37 residue [23].

### 2.6. Circular Dichroism Spectra Analysis

To ascertain the possible effect of the antipsychotics binding on the α-helical secondary structure of HHb, far-UV CD spectroscopy was used. CD measurements in the presence of a fixed concentration of antipsychotics examined (a molar protein to drug ratio of 1:4) are shown in Figure 6A. Consistent with the literature, the CD spectrum of HHb in an aqueous buffer has two characteristic peaks of strong negative ellipticity near 208 nm and 222 nm, ascribed to a typical α-helical protein structure [28]. The negative peak near 208 nm is due to α-helix’s π-π* transition, while the peak near 222 nm is attributed to n-π* transitions for both the α-helix and random coil [29].

The secondary structure of commercial HHb was found to contain 48.2% α-helical regions. The CD spectrum of the HHb showed only slight changes upon adding all three ligands, indicating minor effects of clozapine, ziprasidone, and sertindole binding on the secondary structure of the protein. Still, as shown in Table 3, the α-helical content of HHb borderline significantly (*p* = 0.048) increased to 52.2% in complex with clozapine, while it remained virtually unchanged with ziprasidone and sertindole. The increase in α-helical content suggested a more folded secondary structure and, therefore, a more compact conformation of the HHb molecule in the presence of bound clozapine.

Among other examined drugs, docetaxel, chloroquine, and tramadol were all found to decrease α-helical content in HHb, from 45.3 to 42.8% at the molar ratio of HHb to the ligand of 1:12 [17], from 35.6 to 31.0% at the molar ration 1:6 [19], and from 35.4 to 33.7% at molar ratio of 1:2 [18], respectively. Only diclofenac binding causes the gain of α-helix stability, increasing its content from an initial 41.5% to 49.6% in a 20-fold ligand molar excess [14]. Observed differences in the underestimated content of the α-helices in pure HHb can be mainly attributed to the diversity of used protein preparations. Indeed, in freshly isolated HHb, the calculated α-helix content is found to be ~76%, and it decreased after acetaminophen drug binding [23].

### 2.7. Effect of Atypical Antipsychotics on the Thermal Stability of Human Hemoglobin

Thermal-shift assays are a valuable complement to testing the overall protein stability *per se* and the biochemical consequences of drug actions. The thermal behavior of HHb was studied in the absence and presence of the examined drugs by measuring the unfolding transitions using fluorimetry at a temperature range of 30–85 °C. The relative concentration of protein in each antipsychotic was fixed at a molar ratio of 1:10. The melting temperature of proteins, Tm (midpoint transition temperature), was calculated from thermal denaturation curves (Figure 6B).

The obtained Tm for intact commercial HHb at experiment conditions was 69.3 °C, comparable with the CD results [30]. In the presence of clozapine, Tm was increased to 74.9 °C while remaining effectively unchanged in HHb complexes with ziprasidone and sertindole. Results confirmed the antipsychotic–protein complexing and the stabilizing effect of clozapine binding (CD study results) to higher levels of HHb structure and that ziprasidone and sertindole essentially do not affect (at least) the protein part of its tetrameric structure.

### 2.8. Effect of Atypical Antipsychotics on Spontaneous Human Hemoglobin Oxidation

Hemoglobin spontaneously oxidizes to methemoglobin (metHb) along with the concomitant production of superoxide radicals by a mechanism associated with the distal side of heme, which facilitates the nucleophilic displacement of bound oxygen by the histidine residue [31]. Visible absorption spectroscopy is an effective tool for examining structural changes in colored proteins and their complexation with ligands. Two regions on the normal hemoglobin spectrum can be considered in this part of the electromagnetic spectrum. The first region is the Soret (or B) band, related to heme iron-porphyrin complex absorption at 415 nm. The second region is the Q band, which is associated with the oxy- and deoxy-forms of heme at 550–600 nm. The formation of metHHb is followed by the appearance of a new, small absorption peak with a maximum of around 630 nm [32].

Figure 6C shows the deviation in the Vis absorption spectra of isolated HHb after prolonged incubation with antipsychotics. The calculated metHHb values are presented in Table 3. It was found that both ziprasidone and sertindole in ten-fold molar excess over protein slightly but significantly (*p* < 0.01) increased the amount of met-HHb formed under the applied conditions. This may be because (i) both drugs carry one acid hydrogen atom in their structure, which may be involved in the oxidation process through proton donation and thus enhance the rate of oxidation, or (ii) ziprasidone and sertindole modify the structure of HHb in such a way that it leads to an augmentation of the oxidation rate, or both. The addition of clozapine to the reaction mixture neither significantly promotes nor prevents the formation of oxidized HHb, judging by the identity of the absorption HHb profile in the wavelength range where the met form absorbs and the slightly lower oxidized protein content. The protective effect of clozapine on HHb autoxidation, in comparison with sertindole and especially ziprasidone, may be the consequence of (i) demonstrated (additional) stabilization of the protein’s secondary structure, which slows down the initiation of the autocatalytic stage of the met-HHb formation process, or (ii) antioxidant, including radical scavenging activity of the drug itself [33,34].

Hemoglobin is a polyfunctional molecule involved in functions beyond oxygen transport, such as catalytic activity, nitric oxide metabolism, metabolic reprogramming, redox balance, and pH regulation [35]. Therefore, any substantial change in the active center of the protein (iron protoporphyrin IX prosthetic group) could affect the overall body homeostasis. Considering the low concentrations of ziprasidone (~100 ng/mL) [36] and sertindole (~10 ng/mL) [37] in the plasma of patients on chronic therapy and their relatively low binding constants to highly abundant HHb molecules in vitro that demonstrated a pro-oxidative effect and should not have any significant physiological consequences.

### 2.9. Effect of Atypical Antipsychotics on Oxidant-Assisted Human Hemoglobin Oxidation

Oxidative damage induced by the overproduction and/or accumulation of reactive oxygen species (ROS) can devastate the structure and activity of proteins. Along with by-products of oxygen metabolism, environmental stressors and some xenobiotics (i.e., antineoplastic drugs) greatly increase ROS production [38]. A hydrophilic free radical initiator, AAPH, is a suitable reagent tool for the evaluation of the oxidative stability of bioactive molecules [39]. The ability of antipsychotics to protect the structural elements of isolated HHb from oxidation was examined by monitoring the reduction of the intrinsic fluorescence of the protein induced by AAPH over time. The obtained spectra are shown in Figure 6D, and the calculated protective effect of each drug under the applied experimental setup (corresponding to the difference in areas under the curves in the presence and absence of the drug, expressed in arbitrary units) is depicted as a part of Table 3.

It was obvious that pure HHb is prone to oxidation, considering the fast and substantial decrease of its intrinsic fluorescence intensity, and that the addition of all three antipsychotics in a 10-fold molar excess slowed down this process (Figure 6D). It was found that clozapine delays the AAPH-induced oxidation of HHb molecules the most, followed by ziprasidone and sertindole (Table 3). Therefore, all drugs showed solid antioxidant properties and good protective activity against HHb oxidation.

The biggest increase in the oxidative stability of HHb in the complex with clozapine is not surprising. Its antioxidant properties are well known in various in vitro models [40], and it has shown similar protective effects on plasma proteins fibrinogen [41] and alpha-2-macroglobulin [42]. The suggested therapeutic plasma clozapine concentration (300 to 700 ng/mL) [43] is substantially higher than ziprasidone and sertindole. Therefore, clozapine may actually reduce the oxidation of all these proteins in vivo. All drugs have adverse effects, which can even be potentially life-threatening. There are blood dyscrasias and agranulocytosis/granulocytopenia with clozapine use [44]. Our results point only to the benefits of clozapine complexed with human hemoglobin, without whose evolution life as we know it would not exist.

## 3. Materials and Methods

### 3.1. Materials

Human hemoglobin (lyophilized powder), clozapine, ziprasidone hydrochloride hydrate (purity ≥ 97%), and sertindole (purity ≥ 97.5%) were purchased from Sigma-Aldrich Chemie GmbH (Taufkirchen, Germany) and used without further purification. Unless otherwise stated, HHb solutions were prepared in phosphate-buffered saline (PBS) at pH 7.4, and the protein concentration was determined using Drabkin’s cyanmethemoglobin method [45]. Atypical antipsychotics were dissolved in 99.9% dimethyl sulfoxide (DMSO) to make 4 mM stock solutions. For all experiments, the final concentration of DMSO in various protein-ligand mixtures did not exceed 0.5% (*v/v*). All other chemicals were of analytical reagent grade, and Milli-Q water (Millipore, Molsheim, France) was used in the experiments.

Venous blood was drawn by venipuncture from a healthy volunteer, collected into a tube containing heparin as an anticoagulant, and immediately centrifuged at 2500 *g* for 10 min at 4 °C. Plasma and buffys coat were discarded. The erythrocytes were washed three times with isotonic (0.9%) saline. Packed erythrocytes were lysed in 20 volumes of 10 mM phosphate buffer, pH 7.4 (PB) at 4 °C overnight. Then the hemolysate was centrifuged at 12,000 *g* for 1 h at 4 °C to remove the membrane part. The supernatant hemolysate was collected, and aliquots were applied to size-exclusion chromatography through a Sephadex G-100 column (45 × 1.0 cm), preequilibrated, and eluted with PB. The concentration of obtained HHb was determined from its Soret absorbance at 415 nm (ε = 125,000 M^−1^ cm^−1^) [45]. The purity of the isolated HHb preparations has been checked by 15% SDS-PAGE after staining with Coomassie blue.

### 3.2. Methods

#### 3.2.1. Fluorescence Measurements

All fluorescence data were obtained on a FluoroMax^®^-4 spectrofluorometer (HORIBA Scientific, Kyoto, Japan) under thermostated conditions in a 1.0 cm quartz cell, with the width of the excitation and emission slits both adjusted at 3 or 5 nm. The appropriate blanks, corresponding to the various drugs in buffer concentrations, were subtracted to correct the fluorescence background.

The binding of antipsychotics to HHb was studied by the fluorescence quenching titration method using the intrinsic protein fluorescence probe at constant protein (4 µM) and various ligand concentrations (0 to 28 µM). The steady-state fluorescence spectra were measured at 25, 30, and 37 °C (298.15, 303.15, and 310.15 K, respectively). The excitation wavelength was set at 280 nm, and the emission spectra were read at 290 to 450 nm.

The synchronous fluorescence spectra of the HHb–antipsychotic complexes were recorded at two different scanning intervals: Δλ of 15 nm (tyrosine residue excitation) and Δλ of 60 nm (tryptophan residue excitation), where Δλ = Δλ_EM_ − Δλ_EX_ [24].

The thermal stability of 4 µM commercial human hemoglobin was recorded in the absence and presence of 40 µM of each antipsychotic in the temperature range from 30 to 85 °C. The temperature increment was 2 °C/min, and the equilibration time was adjusted to 1 min. Emission spectra were recorded from 315 to 365 nm, with the excitation wavelength set to 280 nm. The melting curve was presented as the F_350_/F_330_ ratio change with temperature, where F is the emission intensity at a corresponding wavelength, and fitted into a sigmoidal function where the inflection point represents the protein melting point (T_m_) [46].

The stability of proteins towards oxidation was investigated by studying the impact of the strong oxidant 2,2′-azobis(2-amidinopropane) dihydrochloride (AAPH) (final concentration of 5 mM) on the intrinsic purified hemoglobin (4 µM) fluorescence in the absence and presence of drugs at fixed concentrations (40 µM). The fluorescence was monitored during a time course of 30 min at the HHb emission peak of 344 nm after excitation at 280 nm. The protective effect (PE) of each antipsychotic on oxidant-induced reduction of protein fluorescence, expressed in arbitrary units (AU), was calculated using the equation [47]:(1)$$\mathrm{PE}\;\left(\mathrm{AU}\right)\;[\times 10^{6}]={\mathrm{AUC}}_{\mathrm{HHb}+\mathrm{AAP}}\;-\;{\mathrm{AUC}}_{\mathrm{HHb}}$$
where AUC_HHb+AAP_ represents the area under the curve obtained for the human hemoglobin/atypical antipsychotic mixture, and AUC_HHb_ represents the area under the curve obtained for the protein alone.

#### 3.2.2. Circular Dichroism (CD) Measurements

The CD measurements were carried out on a Jasco J-815 spectropolarimeter (Jasco, Tokyo, Japan) in the far-ultraviolet region (185–250 nm) at 37 °C, with two scans averaged for each CD spectrum, using a quartz cell with 0.01 cm path length and 50 nm/min scan speed. The concentration of HHb in PB was 10 µM, and the concentration of drug ligands was 40 µM. The obtained spectra were corrected by subtracting the appropriate spectra from the ligands alone.

The obtained data, presented in units of ellipticity (mdeg), were converted to mean residue ellipticity (MRE) using the following formula:(2)$$\left[\mathrm{MRE}\right]=\frac{\theta \times \mathrm{Mm}}{10\times l\times C\times r}$$
where θ is ellipticity in mdeg, Mm is the molar mass of HHb (64.5 kDa), l is the length of the cuvette, C is the protein molar concentration in the experiment, and r is the number of amino acid residues in the protein (574).

The α-helical hemoglobin content (%) was calculated from these data using the CONTIN algorithm and SP29 database in the CDPro software package.

#### 3.2.3. Visible Absorbance (Vis) Measurements

The absorption spectra of purified human hemoglobin (4 µM) in PB with and without the antipsychotics (40 µM) have been recorded on a UV-1800 spectrophotometer (Shimadzu, Kyoto, Japan) using a 1 cm path length cuvette over a wavelength range from 500 to 700 nm after 6 hrs incubation at 37 °C. The metHHb content (in % of total protein concentration) in samples was determined according to the relation [32]:(3)$$\mathrm{metHHb}=7.0\;\times A_{577}+76.8\;\times \;A_{630}\;-\;13.8\;\times \;A_{560}$$
based on the millimolar extinction coefficients of the tetrameric hemoglobin (oxy, met, and hemichrome) species and the absorbances at the specified wavelengths and pH 7.4.

#### 3.2.4. Statistical Analysis

All experiments were performed in triplicate, and the mean values, standard deviations, and all statistical data were processed using the OriginPro software (OriginLab Corporation, Northampton, MA, USA). Results were analyzed by one-way ANOVA (*p* < 0.05 was considered significant).

#### 3.2.5. Molecular Docking

The initial structure of HHb was obtained from PDB, code 2D60 [48]. Two cocrystallized ligand molecules were removed, and the protein was ionized to pH 7.40 using PROPKA [49]. The protein molecule was then embedded in a 10 Å sphere of TIP3P water molecules and underwent 10 ps of conjugate gradient minimization to eliminate steric bumps while keeping the protein backbone fixed to maintain the experimental structure. The CHARMM force field was employed for this calculation, and NAMD 2.14 [50] was used with Vega ZZ 3.2.0 as the GUI [51].

The 3D structures of clozapine, sertindole, and ziprasidone were retrieved from PubChem, and their final structures were optimized at the PM7 semiempirical level of theory [52] in MOPAC 2016 [53]. The COSMO implicit solvation model of water was utilized by adding the EPS = 78.4 keyword, and the convergence criterion was increased 100 times using the PRECISE keyword.

Molecular docking of three drugs into the HHb was conducted in AutoDock Vina 1.1 [54]. Blind docking was performed by setting the entire protein structure as the receptor site. The exhaustiveness was increased to 200, and 20 binding modes were stored for each ligand.

### 3.3. Theory and Calculations

The fluorescence intensity of a compound decreases (fluorescence quenching) from various molecular interactions, namely, excited state reactions, molecular rearrangements, energy transfer, ground state complex formation, and collisional quenching. To determine the type of quenching, quenching experiments (Section 3.2.1) were performed at three different temperatures (25, 30, and 37 °C), and the data were analyzed by the classical Stern-Volmer equation [15]:(4)$$\frac{F_{0}}{F}=1+K_{q}\;\times \;\tau_{0}\left[Q\right]=1+K_{\mathrm{sv}}\times \left[Q\right]$$
where F_0_ and F are protein emission fluorescence at 344 nm without and with the addition of ligand, respectively, *K*_q_ is the quenching rate constant *τ*_0_ is the average lifetime of the protein in the absence of ligand, which is of the order of 10^−8^ [55]. Further, [Q] is the quencher (ligand) concentration, and *K*_SV_ is the Stern-Volmer dynamic quenching constant, which is obtained from the slope of the linear F_0_/F vs. [Q] plot.

Fluorescence intensities were corrected for the inner-filter effect (the absorption of the exciting light and reabsorption) according to the relationship [15]:(5)$$F_{c}=F_{0}\times 10^{(\mathrm{AEX}+\mathrm{AEM})/2}$$
where F_0_ is measured fluorescence, F_c_ is corrected fluorescence, and A_EX_ and A_EM_ are the absorbances of the quencher at excitation and pick emission wavelengths (344 nm), respectively.

When ligand molecules bind independently to a set of equivalent sites on a macromolecule, the equilibrium between free and bound molecules is given by the following equation [56]:(6)$$\log\frac{F_{0}\;-\;F}{F}=-\;n\;\times \log\frac{1}{\left[L\right]-\left[P\right]\times \frac{F_{0}\;-\;F}{F_{0}}}+n\times \log K_{a}$$
where F_0_ and F are the fluorescence intensities before and after the addition of ligand (an antipsychotic drug), [P] and [L] are the total protein (HHb) and the total ligand concentration, respectively. From the plot of log ((F_0_ − F)/F) vs. log(1/([L] − [P]) × (F_0_ − F)/F)), the binding (association) constant *K*_a_ and the number of binding sites n under a specified set of experimental conditions can be evaluated.

The binding forces contributing to interactions of small organic compounds with proteins frequently include van der Waals interactions, hydrophobic effects, electrostatic interactions, and hydrogen bonds [57]. As the main evidence used to determine the binding mode, the thermodynamic parameters: enthalpy change (Δ*H*), entropy change (ΔS), and Gibbs (free) energy change (ΔG), were calculated from the van’t Hoff equations:(7)$$\ln K_{a}=-\frac{\Delta H}{R}\times \frac{1}{T}+\frac{\Delta S}{R}$$
and the relation:(8)ΔG=ΔH − T×ΔS
where *K*_a_ is the binding constant, *R* is the universal gas constant, and T is the temperature in Kelvins. The values of Δ*H* and Δ*S* were calculated from the slope and intercept of the linear plot of ln*K*_a_ vs. 1/T under the assumption that Δ*H* and Δ*S* are only very weakly dependent on temperature, which is usually valid. Human hemoglobin does not undergo structural degradation at the chosen measurement temperatures (298.15, 303.15, and 310.15 K).

## 4. Conclusions

The interaction models in the system of HHb–antipsychotics clozapine, ziprasidone, and sertindole have been reported for the first time. Different spectroscopic techniques have studied the drug’s binding. The ground-state interactions have been studied by steady-state fluorescence and absorption spectroscopy. The extent of binding and corresponding thermodynamic parameters have also been determined from fluorescence experiments. The structural changes of proteins in the absence and presence of drugs have been studied by circular dichroism and synchronous fluorescence spectra. Theoretical docking analyses have predicted the probable binding site.

All drugs effectively quenched the intrinsic fluorescence of HHb by a static quenching mechanism under simulated pathophysiological conditions. The binding of clozapine, ziprasidone, and sertindole to HHb is unequivocally established, with an affinity of the order of magnitude of 10^3^–10^4^ M^−1^, similar to the values obtained for the binding of most other drugs to this red blood cell protein. From the experimental data, we also conclude that HHb possesses one binding site for the examined antipsychotics. The binding was spontaneous and driven by negative enthalpies and favorable positive entropy contributions. In accordance with the previously described studies concerning the binding interactions of HHb with other small lipophylic heterocyclic drugs, the hydrophobic forces, with the help of electrostatic interactions or hydrogen bonds, play a significant role in stabilizing the HHb–antipsychotic complexes. Molecular docking studies further showed all three drugs bound HHb within its central cavity, consistent with the literature for hydrophobic ligands. Upon interaction, clozapine beneficially influenced the structural integrity properties of tetrameric HHb, based on the increase in α-helical content and the melting point, but reduced oxidant-induced protein deterioration. In contrast, sertindole and ziprasidone showed potentially harmful effects in the complex with HHb, primarily the acceleration of the spontaneous oxidation of an iron atom in the heme structure into the methemoglobin form that does not reversibly bind oxygen.

This work provides a comprehensive understanding of the interactions of well-prescribed second-generation antipsychotics with the physiologically important protein HHb as a biological target. Therefore, our study results may further help understand the pharmacology behavior of these lipophilic drugs and their clinical therapeutic potential, which depends on the drug’s unbound concentration in the blood.

## Figures and Tables

**Figure 1 ijms-24-08921-f001:**
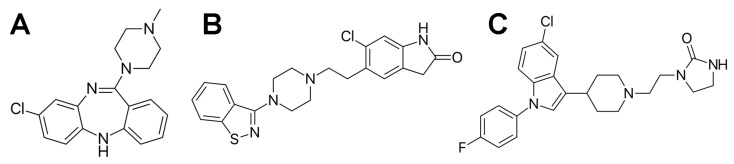
The molecular structure of the tested drugs used in the treatment of schizophrenia. A tricyclic dibenzodiazepine clozapine (3-chloro-6-(4-methylpiperazin-1-yl)-11*H*-benzo[b][1,4]benzo-diazepine) (**A**). A benzothiazolylpiperazine derivative ziprasidone (5-[2-[4-(1,2-benzothiazol-3-yl)piperazin-1-yl]ethyl]-6-chloro-1,3-dihydroindol-2-one) (**B**). A phenylindole derivative sertindole (1-[2-[4-[5-chloro-1-(4-fluorophenyl)indol-3-yl]piperidin-1-yl]ethyl]imidazolidin-2-one) (**C**).

**Figure 2 ijms-24-08921-f002:**
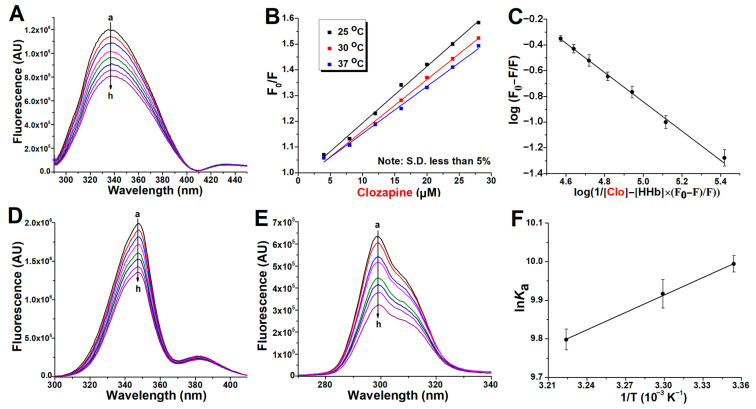
Binding of clozapine (Clo) to commercial human hemoglobin (HHb). Representative emission spectra (excitation at 280 nm) of HHb (4 µM) in the presence of different concentrations of Clo (0, 4, 8, 12, 16, 20, 24, and 28 µM, for curves **a** to **h**, respectively), at 37 °C (**A**). The Stern-Volmer plot of the protein fluorescence quenching at three temperatures (**B**). The plot of the fluorescence quenching data for the estimation of the binding constant and approximation of the number of binding places at 37 °C (**C**). Representative synchronous fluorescence spectra of HHb (4 µM) with (**D**) Δλ of 60 nm (Trp) and with (**E**) Δλ of 60 nm (Tyr) in the presence of increasing Clo concentration (0–28 µM for curves **a** to **h**, respectively). van’t Hoff plot of the binding of Clo with 4 µM HHb for the thermodynamic analysis (**F**). Each data point indicates the average of three determinations for all plots, where error bars indicate the standard deviation.

**Figure 3 ijms-24-08921-f003:**
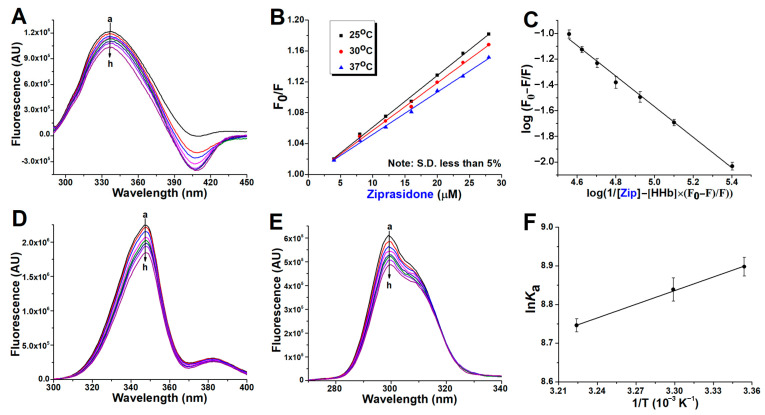
Binding of ziprasidone (Zip) to commercial human hemoglobin (HHb). Representative emission spectra (excitation at 280 nm) of HHb (4 µM) in the presence of different concentrations of Zip (0, 4, 8, 12, 16, 20, 24, and 28 µM, for curves **a** to **h**, respectively), at 37 °C (**A**). The Stern-Volmer plot of the protein fluorescence quenching at three temperatures (**B**). The plot of the fluorescence quenching data for the estimation of the binding constant and approximation of the number of binding places at 37 °C (**C**). Representative synchronous fluorescence spectra of HHb (4 µM) with (**D**) Δλ of 60 nm (Trp) and with (**E**) Δλ of 60 nm (Tyr) in the presence of increasing Zip concentration (0–28 µM for curves **a** to **h**, respectively). (**F**) van’t Hoff plot of the Zip binding with 4 µM HHb for the thermodynamic analysis. Each data point indicates the average of three determinations for all plots, where error bars indicate the standard deviation.

**Figure 4 ijms-24-08921-f004:**
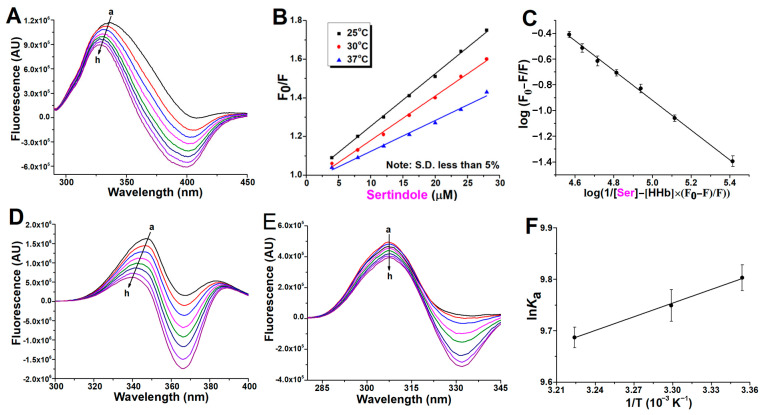
Binding of sertindole (Ser) to human hemoglobin (HHb). Representative emission spectra (excitation at 280 nm) of HHb (4 µM) in the presence of different concentrations of Ser (0, 4, 8, 12, 16, 20, 24, and 28 µM, for curves **a** to **h**, respectively) at 37 °C (**A**). The Stern-Volmer plot of the protein fluorescence quenching at three temperatures (**B**). The plot of the fluorescence quenching data for the estimation of the binding constant and approximation of the number of binding places at 37 °C (**C**). Representative synchronous fluorescence spectra of HHb (4 µM) with (**D**) Δλ of 60 nm (Trp) and with (**E**) Δλ of 60 nm (Tyr) in the presence of increasing Ser concentration (0–28 µM for curves **a** to **h**, respectively). (**F**) van’t Hoff plot of the binding of Ser with 4 µM HHb for the thermodynamic analysis. Each data point indicates the average of three determinations for all plots, where error bars indicate the standard deviation.

**Figure 5 ijms-24-08921-f005:**
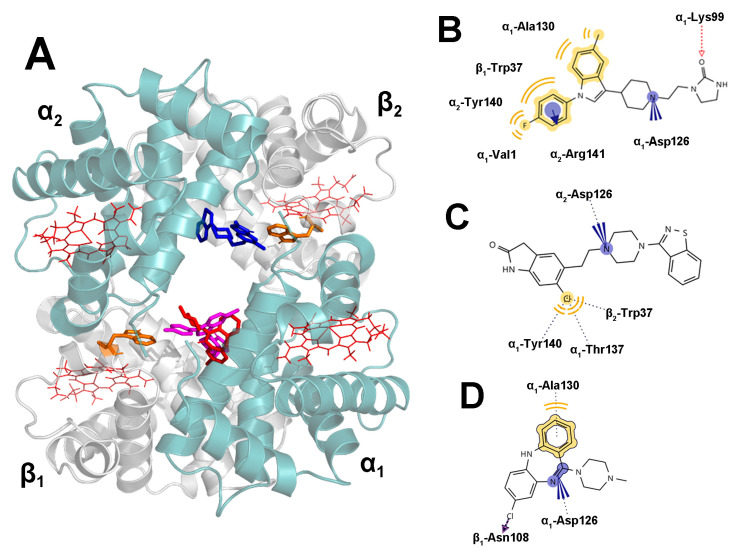
Schematic representation of human hemoglobin (HHb)-antipsychotic docked structures. The lowest-energy binding poses of ziprasidone (blue), clozapine (red), and sertindole (magenta) within the central cavity of HHb (PDB: 2D60); Trp37 residues on β1 and β2 subunits are colored oranges; the image was rendered in PyMOL (**A**). The 2D ligand-interaction diagrams for HHb with sertindole (**B**), ziprasidone (**C**), or clozapine (**D**); images were prepared in LigandScout 4.4.

**Figure 6 ijms-24-08921-f006:**
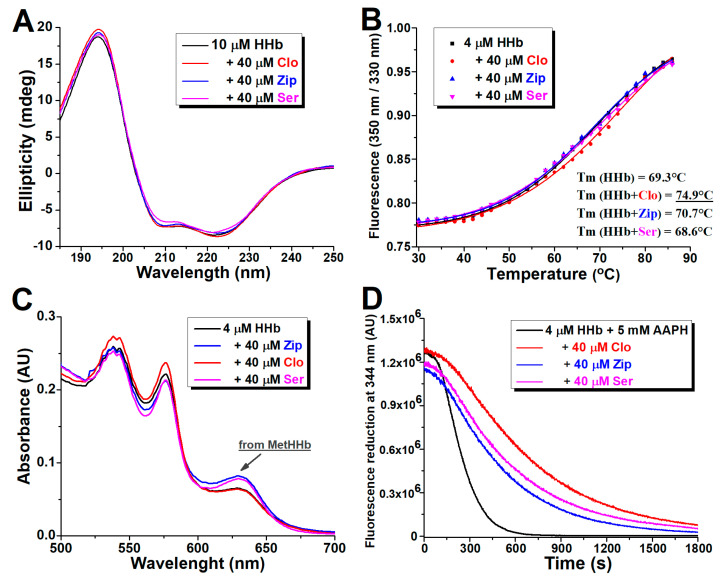
Binding effects of antipsychotics clozapine (Clo), ziprasidone (Zip), and sertindole (Ser) on the structure and oxidative stability of human hemoglobin (HHb). Representative far UV-CD spectra of commercial HHb in the absence and presence of tested drugs (**A**). Representative melting curves of commercial HHb without or with antipsychotic addition, with the specified mean values of the obtained protein melting points (Tm) (**B**); the standard deviations were less than 5% of the means. Representative visible absorption spectra of isolated HHb, alone and with the ligands (**C**). Representative changes in the isolated HHb emission intensity induced by 2,2′-azobis(2-amidinopropane) dihydrochloride (AAPH), without and in the presence of tested drugs (**D**).

**Table 1 ijms-24-08921-t001:** Quenching parameters of the interaction of atypical antipsychotics with human hemoglobin at three different temperatures.

Ligand	T (K)	*K*_SV_ (M^−1^)	R^2^	*K*_q_ (M^−1^ s^−1^)
	298.15	2.22 × 10^3^	0.9958	2.22 × 10^11^
Clozapine	303.15	2.01 × 10^3^	0.9957	2.01 × 10^11^
	310.15	1.84 × 10^3^	0.9952	1.84 × 10^11^
	298.15	0.67 × 10^3^	0.9956	0.67 × 10^11^
Ziprasidone	303.15	0.62 × 10^3^	0.9957	0.62 × 10^11^
	310.15	0.55 × 10^3^	0.9970	0.55 × 10^11^
	298.15	2.71 × 10^3^	0.9981	2.71 × 10^11^
Sertindole	303.15	2.29 × 10^3^	0.9957	2.29 × 10^11^
	310.15	1.16 × 10^3^	0.9911	1.16 × 10^11^

*K*_sv_, the Stern-Volmer (SV) quenching constant; R^2^, linear correlated coefficient of SV plots; *K*_q_, the bimolecular quenching constant. The standard deviation for all parameters was less than 5% of the mean.

**Table 2 ijms-24-08921-t002:** Binding and thermodynamic parameters of human hemoglobin-antipsychotic complexes at three different temperatures.

Ligand	T (K)	*K*_a_ × 10^4^ (M^−1^)	n	Δ*G* (kJ mol^−1^)	Δ*S* (J mol^−1^ K^−1^)	Δ*H* (kJ mol^−1^)
	298.15	2.19	1.12	−12.2		
Clozapine	303.15	2.03	1.03	−12.4	+41.0	−12.5
	310.15	1.80	0.97	−12.7		
	298.15	0.74	1.16	−12.3		
Ziprasidone	303.15	0.69	1.08	−12.5	+41.2	−9.8
	310.15	0.63	1.06	−12.8		
	298.15	1.81	1.10	−16.9		
Sertindole	303.15	1.71	1.06	−17.2	+56.8	−7.4
	310.15	1.61	1.03	−17.6		

*K*_a_, the binding (association) constant; n, the number of binding sites on the protein; Δ*H* and Δ*S* are the enthalpy and entropy changes, respectively; Δ*G*, the free energy change. The standard deviation for all parameters was less than 5% of the mean, and the linearly correlated coefficients of plots for *K*_a_ and n determination (Figure 2C, Figure 3C and Figure 4C) were higher than 0.99.

**Table 3 ijms-24-08921-t003:** The effects of human hemoglobin (HHb)–antipsychotic complexes formation on selected structural-related protein parameters.

Sample & Monitored Parameter	α-Helices Content (%)	metHHb Content (%)	Protection from Oxidation (AU)
Control HHb (4 µM)	48.2 ± 0.7	3.96 ± 0.20	1 (reference)
+clozapine (10 µM)	52.2 ± 0.5 ^a^	3.86 ± 0.23	118 ± 4.1
+ziprasidone (10 µM)	50.2 ± 0.6	5.32 ± 0.27 ^c^	78 ± 3.2
+sertindole (10 µM)	49.9 ± 1.0	5.14 ± 0.42 ^b^	60 ± 2.8

Significantly different from the control values: ^a^—*p* < 0.05; ^b^—*p* < 0.01; ^c^—*p* < 0.001. MetHHb: oxidized, ferrihemoglobin; AU: arbitrary units.

## Data Availability

Not applicable.

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
