# Peer review of "Human Hemoglobin and Antipsychotics Clozapine, Ziprasidone and Sertindole: Friends or Foes?"

_ijms, 2023, doi:10.3390/ijms24108921_

Round 1

Reviewer 1 Report

In this study, human erythrocytes are a model system for testing the pleiotropic effects of lipophilic drugs in order to study the interaction between the antipsychotics clozapine, ziprasidone, sertindole, and human hemoglobin. This is a very relevant and interesting topic. I read that this article is suitable for publication, it will be of interest to a wide range of researchers (pharmacologists, toxicologists, psychiatrists and neurologists, diochemists, medical chemists) and, therefore, will be well cited. I recommend a minor revision to improve the quality of the article. Explain in detail the need to study the effect of atypical antipsychotics on the thermal stability of human hemo-globin Binding and thermodynamic parameters of human hemoglobin–antipsychotic complexes 246 at three different temperatures: justify the choice of temperature range Discuss the possibility of other approaches to binding constants determination in human hemoglobin and antipsychotic systems. After all, there may be different results. How confident are you in the correctness of the chosen approach? The introduction section should be shortened. The introduction should state the hypothesis of your research.

The quality of English language is fine.

Reviewer 2 Report

The manuscript “Human hemoglobin and antipsychotics clozapine, ziprasidone and sertindole: friends or foes?” by Lena Platanić Arizanović et al. has employed in vitro and in silico tools to investigate the binding of three antipsychotic drugs clozapine, ziprasidone, and sertindole to human hemoglobin (HHb) and evaluate their pleiotropic effects. Overall, I find this is a well-organized manuscript and it merits publication in Int. J. Mol. Sci. However, there are a couple of concerns for the authors to address before publication.

1. It reads to me that the authors aim to inspect the pleiotropic effects of the three antipsychotic drugs, which may act on other targets for which they were not specifically developed. However, I am unclear why the authors selected HHb as their target. Have there been any prior biochemical studies or clinical reports indicating that these drugs could impact HHb, either adversely or beneficially?

In other words, the authors have not justified the necessity of their study. The authors should revise at least the Introduction to clarify why it is important to study the pleiotropic effects of these antipsychotic drugs on HHb.

2. The fluorescence has shown that sertindole binds stronger to HHb than clozapine and ziprasidone by about 1 kcal/mol. However, the molecular docking has given the opposite result. This does not surprise me because the docking affinities do not include the entropic contributions, hence they align with the binding enthalpies obtained from the fluorescence experiments. Nonetheless, the authors have not failed to pointed this out. In addition, it is worth noting that the strongest binding of sertindole to HHb, as determined from the fluorescence, is attributed to the significant entropic effects. I am glad to see that the authors have acknowledged this, but I cannot agree with the explanation that “the positive entropy changes meant that the hydrophobic forces played a significant role during the binding” (page 7, lines 257–259). I don’t understand the reasoning behind this statement. A large binding entropy typically indicates considerate conformational rearrangements of the ligand and/or target during the binding.
